# Classification of Building Damage Using a Novel Convolutional Neural Network Based on Post-Disaster Aerial Images

**DOI:** 10.3390/s22155920

**Published:** 2022-08-08

**Authors:** Zhonghua Hong, Hongzheng Zhong, Haiyan Pan, Jun Liu, Ruyan Zhou, Yun Zhang, Yanling Han, Jing Wang, Shuhu Yang, Changyue Zhong

**Affiliations:** 1College of Information Technology, Shanghai Ocean University, Shanghai 201306, China; 2National Earthquake Response Support Service, Beijing 100049, China; 3College of Civil Engineering and Architecture, Guizhou Minzu University, Guiyang 550025, China

**Keywords:** building damage, deep learning, earthquake building damage classification net (EBDC-Net), aerial images

## Abstract

The accurate and timely identification of the degree of building damage is critical for disaster emergency response and loss assessment. Although many methods have been proposed, most of them divide damaged buildings into two categories—intact and damaged—which is insufficient to meet practical needs. To address this issue, we present a novel convolutional neural network—namely, the earthquake building damage classification net (EBDC-Net)—for assessment of building damage based on post-disaster aerial images. The proposed network comprises two components: a feature extraction encoder module, and a damage classification module. The feature extraction encoder module is employed to extract semantic information on building damage and enhance the ability to distinguish between different damage levels, while the classification module improves accuracy by combining global and contextual features. The performance of EBDC-Net was evaluated using a public dataset, and a large-scale damage assessment was performed using a dataset of post-earthquake unmanned aerial vehicle (UAV) images. The results of the experiments indicate that this approach can accurately classify buildings with different damage levels. The overall classification accuracy was 94.44%, 85.53%, and 77.49% when the damage to the buildings was divided into two, three, and four categories, respectively.

## 1. Introduction

As some of the most catastrophic events in nature, earthquakes can cause significant structural damage to buildings [1]. The timely and accurate classification of the degree of building damage is of great importance to the government’s emergency response and rescue operations. Remote sensing images can be used to obtain abundant spatiotemporal information in the affected area so that buildings can be evaluated on a large scale, at low cost, and quickly [2].

Convolutional neural networks (CNNs) have powerful feature learning and inference capabilities, as well as strong performance in image processing tasks [3]. Therefore, CNNs are widely used in the damage assessment of buildings. According to the number of images used, building damage assessment methods are classified as dual-temporal and single-temporal methods [4].

The dual-temporal methods extract features from the pre- and post-disaster images, and then determine the localization and the degree of damaged buildings [4]. Wu et al. constructed a Siamese neural network with different backbones to automatically detect damaged buildings, and used the attention gate to filter useless features [5]. Xiao et al. proposed a dynamic cross-fusion network that enables the localization and classification tasks of buildings to share feature information from different levels of the CNN network, enhancing information exchange across tasks [6]. Adriano et al. developed a damage assessment network by combining multimodal and multi-temporal data to increase the utility of the model under different data sources [7]. Since the dual-temporal methods employ both pre- and post-disaster images, they usually have a high classification accuracy [8]. However, the practicality of these methods is greatly limited due to the accessibility of dual-temporal images [9,10].

Single-temporal methods only use post-disaster images for building damage assessment tasks. Therefore, they are subject to relatively few constraints. Duarte et al. proposed a CNN with multiresolution feature fusion to increase the performance of the model in multiresolution images [11]. Ji et al. explored the use of pre-trained CNN and fine-tuned CNN strategies for the damage classification of buildings after earthquakes [12]. Nex et al. assessed the migration performance of the CNN using images of different locations and spatial resolutions [13]. Ishraq et al. replaced the fully connected layer in the CNN with a global average pooling layer to assess building damage caused by hurricanes [14]. However, the majority of the existing research divides buildings into two categories—intact and damaged—which cannot meet the needs of rescue and post-disaster damage refinement assessment. More detailed classification information about the degree of damage to buildings is needed [15]. Ci et al. combined a CNN with ordinal regression to classify building damage as intact, slightly damaged, severely damaged, or collapsed [16]. Ma et al. used geographic information system (GIS) data to provide evident boundary characteristics of buildings. A CNN model combined with GIS data was proposed to classify building damage into slight damage, moderate damage, and severe damage [17]. However, these studies ignore the impact of the extraction of building damage features on classification accuracy. In post-disaster images, the shape and texture of the building change significantly [18]. Distinguishing between slight damage and severe damage is a challenging task because they share similar characteristics. For instance, the damage characteristics of buildings are mainly manifested in roofs, except that the damaged area is different. Therefore, it is necessary to aggregate similar features to enhance the discrimination of different degrees of building damage [2]. In addition, texture and spatial information around the buildings can provide necessary auxiliary information for evaluation. Exploring the relationship between global features and context features in images helps the model to classify the damage levels of buildings more accurately.

To address the abovementioned issues, this study proposes a novel CNN—namely, the earthquake building damage classification net (EBDC-Net)—for assessment of building damage using post-disaster aerial images. The proposed network is made up of a feature extraction encoder module and a damage classification module. The feature extraction encoder module is used to extract the semantic information and enhance the feature representation capability of different damage levels of buildings from the images, while the damage classification module is used to fuse the global and contextual features to improve the accuracy of damage classification.

The rest of this paper is organized as follows: Section 2 introduces the data sources and the proposed method. The experimental results are presented in Section 3. Section 4 discusses the role of historical earthquake data in new earthquakes. Finally, some conclusions are drawn in Section 5.

## 2. Materials and Methods

### 2.1. Data Sources

The datasets used in this study contain post-earthquake images from three different locations. The first comprises post-earthquake aerial images of the 7.1 magnitude earthquake that occurred on 14 April 2010 in Yushu County, Qinghai Province, China [16]. The second comprises post-earthquake aerial images of the 6.5 magnitude earthquake that occurred on 3 August 2014 in Ludian County, Yunnan Province, China [16]. The third comprises post-earthquake UAV images of the 6.4 magnitude earthquake that occurred on 21 May 2021 in Yangbi County, Yunnan Province, China. The Yushu and Ludian datasets are public datasets, and can be downloaded from https://github.com/city292/build_assessment (accessed on 1 April 2022) [16]. For the Yangbi dataset, the region of interest (ROI) was first cropped from the post-disaster UAV images of the Yangbi earthquake. Then, the ROI was cropped into patches of different sizes according to the resolution of the images and the structural features of the local buildings. Finally, all patches were uniformly resized to 88 × 88 pixels. As shown in Table 1, the datasets classify building damage into four categories: intact, slightly damaged, severely damaged, and collapsed. As shown in Table 2, the number of images with four damage levels in the dataset was counted.

To verify the performance of EBDC-Net on the different classification criteria, the buildings with different damage levels were divided into three groups, as shown in Table 3. Group 1 simply divided all of the buildings into non-collapsed and collapsed, without distinguishing the degrees of damage to the buildings. Group 2 contained three categories, namely, intact, severely damaged, and collapsed. Slightly damaged buildings were considered to be intact. For Group 3, a more detailed classification criterion was devised, and all of the buildings were divided into four categories: intact, slightly damaged, severely damaged, and collapsed.

### 2.2. Methods

As shown in Figure 1, a building damage classification framework—namely, EBDC-Net—is proposed in this study. EBDC-Net is composed of a feature extraction encoder module and a building damage classification module. First, a building damage feature extraction encoder module was constructed to extract the semantic information of different damage levels. In feature extraction, the spatial attention mechanism (SAM) is used to gather similar features in the image to enhance the feature representation ability of the network. Second, two parallel modules—global feature extraction (GFE) and contextual feature extraction (CFE)—were included in the building damage classification module to fully exploit the global features and contextual features in the images for building damage classification.

#### 2.2.1. Feature Extraction Encoder Module

Extracting useful features from the post-disaster aerial images helps to determine the degree of damage to buildings [19]. As shown in Figure 1, the post-disaster aerial images were used as input data for EBDC-Net. The encoder module of EBDC-Net consists of four convolutional blocks stacked together. In blocks 1 and 2, each convolutional block contains two 2D convolutions (kernel = 3). In blocks 3 and 4, each convolutional block contains three 2D convolutions (kernel = 3). Both convolution operations were followed by a maximum-pooling downsampling operation (kernel = 2). In addition, due to the small size of the input images in this study, there was significant feature loss as the network deepened. The structure of the residual connections can reduce the difficulty of optimization, and enables the training of deeper networks [20]. Therefore, residual connections were added to the last three convolution blocks to alleviate the gradient disappearance and gradient explosion problems during feature extraction.

Although convolutional blocks can extract the semantic information of different levels, in the post-disaster aerial images, the damage features of slightly and severely damaged buildings were scattered in different areas of the images, and the proportion of damage features in the images was small, resulting in a large intraclass variance between different damage categories in the same image. To enhance the feature representation ability of the encoder module and obtain better classification accuracy, a spatial attention mechanism (SAM) was introduced at the end of each convolutional block [21]. The SAM adaptively explores similarities between features at different locations in the image, integrates similar features at any scale, increases intraclass consistency between different damage classes, and suppresses unwanted information and noise.

As shown in Figure 2, the SAM received the feature maps $F_{A}\in R^{C\times H\times W}$ extracted from the convolutional block, where  C, H, and W represent the channel, height, and width of $F_{A}$, respectively. First, $F_{A}$ was used as an input, and two new feature maps$—F_{B}$ and $F_{C}$—were obtained through two convolutional layers (kernel = 1). The output channels of these two convolutional layers were C/8. Second, $F_{B}$ was reshaped as $R^{N\times \left(\frac{C}{8}\right)}$, and $F_{C}$ was reshaped as $R^{\left(\frac{C}{8}\right)\times N}$, where N=H×W. Subsequently, $F_{B}$ and $F_{C}$ underwent matrix multiplication to generate the feature map $F_{S^{\prime }}\in R^{N\times N}$. Finally, $F_{S}$ was fed into the softmax layer, and the attention weight map $F_{S}\in R^{N\times N}$ was generated.
(1)$$F_{S}{}_{ji}=\frac{\exp\left(F_{B}{}_{i}\cdot F_{C}{}_{j}\right)}{\sum_{i=1}^{N}\exp(F_{B}{}_{i}\cdot F_{C}{}_{j})}$$

$F_{S}{}_{ji}$ was used to measure the influence between the features of any two positions in the space. The closer the representation of the features of two positions, the stronger the correlation between them, and the larger the value of $F_{S}{}_{ji}$. After obtaining the attention weight map, $F_{A}$ was fed into a new convolution layer to generate a new feature map $F_{D}$. $F_{D}$ and $F_{A}$ had the same shape. A matrix was multiplied between the sum space of the attention weight feature map $F_{S}$ and the feature map $F_{D}$, and the result was reshaped as $R^{C\times H\times W}$. Finally, this result was multiplied by a trainable scale factor α and summed with $F_{A}$ to obtain the final feature map $F_{E}$, with α initialized to 0.
(2)$$F_{E_{j}}=SAM\left(F_{A}\right)=\alpha \;\overset{N}{\underset{i=1}{\sum }}F_{S}{}_{ji}F_{D_{i}}+F_{A_{j}}$$

According to Equation (2), the value of each position of $F_{E_{j}}$ was obtained through the weighted fusion of the original features, with the values in $F_{S}$ as weights. Therefore, SAM selectively aggregated similar semantic features to improve intraclass compactness and semantic consistency between different damage classes, enabling the network to better distinguish between buildings of different damage classes.

#### 2.2.2. Building Damage Classification Module

When using post-disaster aerial images to classify building damage levels, focusing only on the characteristics of the building itself is not enough to accurately distinguish its damage level. Scenes around buildings can provide necessary auxiliary information for damage assessment. If the global information and contextual dependencies in the images are taken into account, it may improve the final damage classification results. As shown in Figure 1, two parallel modules were designed in the building damage classification module to capture the global information and contextual feature dependencies in the images, respectively. The feature F extracted by the feature extraction encoder module was used as the input to the building damage classification network. Specifically, F was first fed into the GFE module, where the global feature vector $F_{G}$ of the image was extracted using a global-level pooling layer. Then, F was fed into the CFE module. In the CFE module, the long short-term memory (LSTM) layer [22] was used to extract the contextual feature dependencies $F_{C}$ in the image.

As a deep regression neural network, LSTM can handle long-term relationships of memory sequence information [22]. Many remote sensing image classification studies use LSTM to extract spatial and spectral features from images [23,24,25]. In this study, LSTM was used to explore the contextual dependencies between different regional feature sequences. The generation of feature sequences is the key to learning contextual feature dependencies. As shown in Figure 3, $F\in R^{C\times H\times W}$ was transformed into a feature sequence $V=\left[x_{1},\;x_{2},\;\cdot \cdot \cdot ,\;x_{K}\right]\in R^{C}$, where K=H×W. Each C-dimensional feature vector $x_{k}$ was fed into the LSTM sequentially as a feature sequence.

As shown in Figure 4, LSTM has three inputs: the input value $x_{k}$ of the current feature sequence, the output value $h_{k-1}$ of the previous feature sequence, and its cell state $C_{k-1}$. LSTM has two outputs: the output value $h_{k}$ of the current feature sequence, and its cell state *C_k_*. The forgetting gate $f_{k}$ combines $h_{k-1}$ and $x_{k}$ to determine how much of the cell state $C_{k-1}$ of the previous feature sequence is retained in the current feature sequence. The input gate $i_{k}$ combines $h_{k-1}$ and $x_{k}$ to determine how much of the input ${\tilde{C}}_{k}$ of the current feature sequence is preserved in the new cell state $C_{k}$. The output gate $o_{k}$ combines the cell state $C_{k}$ of the current feature sequence to control the current output value $h_{k}$. Based on these components, the storage cells and their outputs can be computed as follows:(3)$$f_{k}=\sigma \left(W_{f}\left[h_{k-1},x_{k}\right]+b_{f}\right)$$
(4)$$i_{k}=\sigma \left(W_{i}\left[h_{k-1},x_{k}\right]+b_{i}\right)$$
(5)$${\tilde{C}}_{k}=\tanh\left(W_{C}\left[h_{k-1},x_{k}\right]+b_{C}\right)$$
(6)$$C_{k}=f_{k}\;\circ \;C_{k-1}+i_{k}\circ {\tilde{C}}_{k}$$
(7)$$o_{k}=\sigma \left(W_{o}\left[h_{k-1},x_{k}\right]+b_{o}\right)$$
(8)$$h_{k}=o_{k}\circ tanh\left(C_{k}\right)$$
where σ represents the sigmoid function, ′∘′ is the Hadamard product, and $W_{f}$, $W_{i}$, $W_{C},\;W_{0}$, $b_{f}$, $b_{i}$, $b_{C}$, and $b_{o}$ are learnable weights. The output state of the last feature sequence was used as the contextual feature $F_{C}$ to describe the contextual feature dependencies in the image. In the contextual feature extraction module, two LSTM layers were stacked, and the output dimensions were set to 256.

After obtaining the global feature $F_{G}$ and the contextual feature $F_{C}$, the two features were fed into the concatenate layer for effective connection to obtain the fusion feature $F_{fusion}$. Then, $F_{fusion}$ was fed into two fully connected layers of 256 dimensions. It is worth noting that the dropout strategy was used to avoid overfitting. Finally, the features were processed using the softmax function to output the damage level of the building.
(9)$$F_{fusion}=\left[F_{G},\;F_{C}\right]$$

## 3. Results

### 3.1. Implementation Details of the Experiment

All of the experiments and tests in this study were conducted on the same platform, configured with 32 GB RAM, an i7 9800X @3.8 GHz CPU, and a GeForce RTX 2080 Ti GPU. The ratio of the training set, validation set, and test set in the Ludian dataset was 8:1:1, respectively. Since the numbers of images in the Yushu and Yangbi datasets are smaller, the ratio of the training set, validation set, and test set was 6:2:2, respectively.

To obtain the optimal hyperparameters, different batch sizes and learning rates were tested individually, where one of the hyperparameters was fixed.

As shown in Table 4 and Table 5, the highest accuracy of the model was achieved when the learning rate was 0.0001 and the batch size was 32. Meanwhile, as shown in Figure 5, the model converged when it was trained for 100 epochs.

The model was trained using the SGD optimizer; the weight decay was 0.001, and the cross-entropy loss function was used. The pre-training weight on ImageNet was used to initialize the feature extraction encoder network. Horizontal and vertical flips were used for data enhancement during training. To ensure fairness, the parameters of all of the comparison methods were the same as those of the proposed method. In this study, the overall accuracy (OA), kappa coefficient, and mean square error (MSE) were used as indicators to evaluate the classification accuracy of the model.

### 3.2. Results of the Comparison of Different Baseline Models

We first compared the performance of different baseline models in the Ludian and Yushu datasets. The classification results for the Yangbi dataset are not presented because there were no collapsed buildings, meaning that it was not possible to divide the buildings into three groups according to the aforementioned grouping criteria. Therefore, the Yangbi dataset was used to discuss the performance of the fine-tuned model. The baseline was constructed by removing the residual connections, SAM module, and CFE module from EBDC-Net. The performance of seven baseline models was compared in Group 1, Group 2, and Group 3, including DenseNet [26], ResNet50 [20], InceptionV3 [27], Xception [28], MobileNet [29], VGG16 [30], and baseline. Table 6 and Table 7 show the quantitative comparison of the building damage classification accuracy of different baseline models in the Ludian and Yushu datasets, respectively. As can be seen from Table 6, all of the seven baselines exhibited similar performance for Group 1, with an overall accuracy higher than 90%, due to its relatively simple classification criterion. However, with the increase in the number of building damage categories, the differences in performance between the different models became greater—especially for Group 3, where the classification accuracy dropped dramatically. Among all of the baseline models, the baseline used in this study performed the best on OA, kappa, and MSE in the three groups of the Ludian dataset.

A similar conclusion can be drawn in the Yushu dataset. As shown in Table 7, for Group 1, all of the baseline models exhibited excellent performance, with an overall accuracy higher than 90%. VGG16 showed the best OA in Group 1, which was 0.42% higher than that of the adopted baseline model. For Groups 2 and 3, the best OA was obtained using the adopted baseline model, which was 0.85% and 1.28% higher than that of VGG-16.

### 3.3. Results of Ablation Experiments

In this paper, ablation experiments were performed to demonstrate the contribution of different modules in EBDC-Net to the classification of building damage, where R represents the residual connections, S represents the SAM module, and C represents the CFE module. Table 8 and Table 9 show the comparison of the ablation experiments in the Ludian and Yushu datasets, respectively. Compared with the baseline, when the residual connections, SAM module, and CFE module were all added to the model, it showed the highest overall accuracy for the three groups of the two datasets. Compared with the baseline, the OA of EBDC-Net improved by 1.05% and 1.42% for Group 1, 2.01% and 1.99% for Group 2, and 3.26% and 2.43% for Group 3, in the Ludian and Yushu datasets, respectively.

These results indicate that EBDC-Net showed more significant advantages in building damage classification tasks where the categories were more finely divided. This is because the residual connections mitigated the loss of small features as the network deepened. Second, SAM enhanced the representation of damage features in the images, and improved the network’s ability to distinguish between intermediate damage classes. Finally, combining global and contextual features of the images improved the classification accuracy. Figure 6 is the confusion matrix between the baseline and EBDC-Net in the Ludian and Yushu datasets. It can be concluded that EBDC-Net is better able to distinguish between buildings with different levels of damage. Thus, EBDC-Net helps in the fine-grained assessment of building damage.

### 3.4. Results of Comparison with Different Building Damage Classification Methods

To verify the effectiveness of EBDC-Net in the classification of building damage, we compared EBDC-Net with four different building damage classification methods. Res-CNN is a model constructed using the CBR module and residual connection [11]. Dense-CNN is a CNN model constructed with dense blocks [13]. The full connection layer in VGG-GAP is replaced by the global average pooling layer [14]. VGG-OR combines the CNN with ordinal regression [16]. As shown in Table 10 and Table 11, the EBDC-Net framework proposed in this study showed the best performance in all three groups. The OA was 94.44% and 94.72% in Group 1, 85.33% and 79.02% in Group 2, and 77.49% and 67.62% in Group 3, respectively. Compared to the other four methods, EBDC-Net had a more significant advantage over Group 3 than Groups 1 and 2, with an overall accuracy of 13.5% and 9.42% higher than Re-CNN, 8.34% and 8.13% higher than Dense-CNN, 4.22% and 3.9% higher than VGG-GAP, and 1.92% and 2.57% higher than VGG-OR, respectively.

As shown in Table 12, there was a small amount of debris around the intact buildings in the first and second images, while the buildings in the seventh and eighth images were buried by large debris, and none of their roofs showed significant damage. EBDC-Net enhanced the model’s ability to distinguish between texture information and spatial structure around the buildings by combining global and contextual features. The third and sixth images correspond to slightly damaged and severely damaged buildings, respectively. In both damage classes, the main body of the building was intact, and the damage to the building was scattered across the roof. SAM can aggregate similar features in images, enhancing the network’s feature representation, and helping to distinguish buildings in intermediate damage categories.

## 4. Discussion

In the building damage classification task, the model learned the damage characteristics of buildings from historical earthquake data. After the earthquake, fine-tuning the model with new data helped to quickly and accurately assess the building damage levels. In this study, three experiments were designed to explore the role of historical data in the post-earthquake building damage assessment task. In Test 1, EBDC-Net was trained using the Ludian dataset and predicted in the Yushu dataset. In Test 2, EBDC-Net was trained and predicted in the Yushu dataset. In Test 3, the Yushu dataset was used to fine-tune EBDC-Net, which was trained in the Ludian dataset.

As shown in Table 13, for Test 1, the OA of the model was 88.30% in Group 1, 69.19% in Group 2, and 56.63% in Group 3. Compared with the results in the Ludian dataset, the classification accuracy of the model in Groups 2 and 3 decreased sharply. As shown in Figure 7, the structure, shape, and style of buildings in the two datasets were very different. The features learned by the model from the Ludian dataset were not enough to represent the features of damaged buildings in the Yushu dataset, leading to low classification accuracy.

Similarly, in Test 2, the OA of the model in Group 2 and Group 3 was 79.02% and 67.62%, respectively, which was lower than the corresponding accuracy in the Ludian dataset. The reason for this is that there were much smaller scales of Yushu dataset images than Ludian dataset images. It was also shown that in the refined assessment of building damage, the number of samples can have a significant impact on the accuracy of the assessment.

However, the accuracy of the model was improved dramatically when the network trained using Ludian images was fine-tuned using a small additional amount of Yushu images. As shown in Table 11, the OA was 95.86%, 80.82%, and 68.33% for the three groups, respectively, which was 7.56% and 1.14% higher for Group 1, 10.83% and 1% for Group 2, and 11.7% and 0.71% for Group 3, compared to Tests 1 and 2, respectively. This is because the historical earthquake data can provide the basic features of the damaged buildings. By adding a small number of images from the testing area, more detailed and local features can be learned, bringing about the improvement of classification accuracy. This indicates that fine-tuning is an effective strategy for the classification of building damage

Through the visual qualitative analysis of the prediction results of some areas, we can intuitively understand the model through the assessment of building damage. In this study, we divided the images into patches, rather than segmentations of individual buildings. Therefore, a patch may contain several buildings. When a building was cropped into two or more patches, the damaged features of the building were retained in the corresponding patches. The EBDC-Net model trained using the Ludian dataset was fine-tuned using the Yangbi dataset. Figure 8 shows the visualization results of the model evaluation and the visual interpretation results (ground truth). The model evaluation results were generally consistent with the visual interpretation results, with an overall accuracy of 75%, a kappa of 0.66, and an MSE of 0.26. In addition, the time needed for UAV image evaluation was tested. The results show that the average processing time for an image of 5474 × 3648 pixels was 19 s.

## 5. Conclusions

In this work, we propose a novel network called EBDC-Net to solve the finer classification problem of damaged buildings after earthquakes. The proposed method was tested using two datasets and compared with four state-of-the-art methods. In addition, the roles of the residual connection, spatial attention mechanism, and contextual feature extraction module were also explored. The experimental results demonstrated the following: (1) in the Ludian and Yushu datasets, the accuracy of the proposed method was at least 1.92% and 2.57% higher compared to state-of-the-art building damage classification methods; (2) with the introduction of the above three strategies, the classification accuracy was improved by 3.26% and 2.43% in the Ludian and Yushu datasets, respectively, compared to the baseline model; and (3) using the historical earthquake data and the fine-tuned model is a good strategy to quickly classify the buildings damaged in the new earthquake. 

The main contributions of this paper can be summarized as follows:(1)We propose a novel deep-learning-based model to solve the fine-grained classification problem of damaged buildings, which is critical to earthquake rescue and post-disaster damage assessment.(2)The spatial attention mechanism and the contextual feature extraction module are embedded in EBDC-Net, which can improve the model’s ability to classify buildings with different levels of damage.

In the future, we will try to explore the classification of building damage under complex conditions through the use of multimodal and multi-temporal remote sensing images.

## Figures and Tables

**Figure 1 sensors-22-05920-f001:**
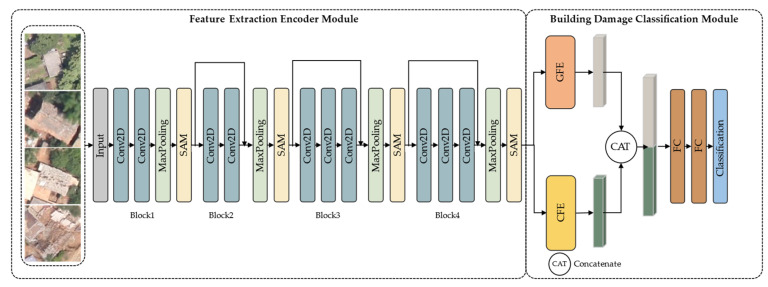
The framework of the proposed EBDC-Net.

**Figure 2 sensors-22-05920-f002:**
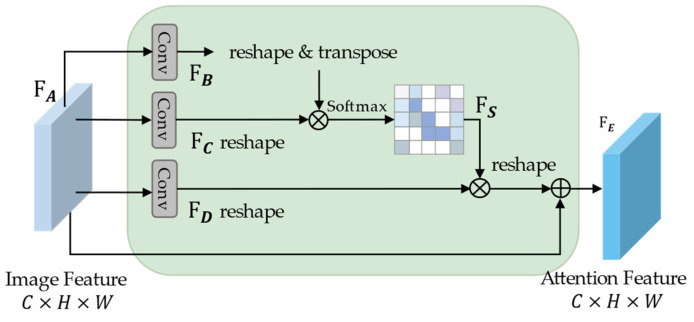
Spatial attention mechanism.

**Figure 3 sensors-22-05920-f003:**
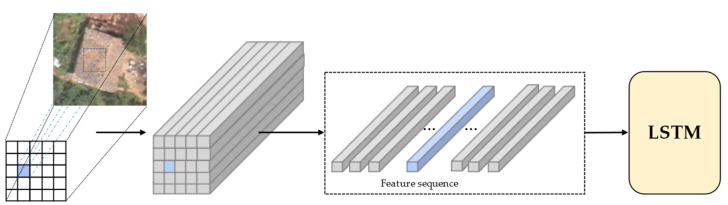
Feature sequence generation.

**Figure 4 sensors-22-05920-f004:**
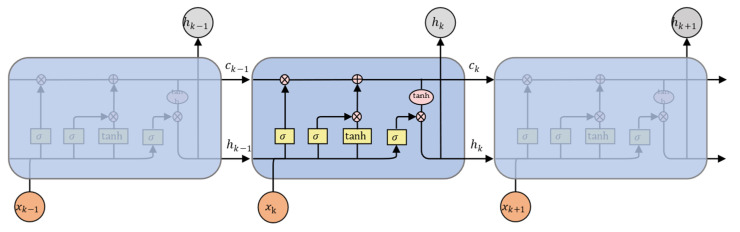
LSTM processing cell.

**Figure 5 sensors-22-05920-f005:**
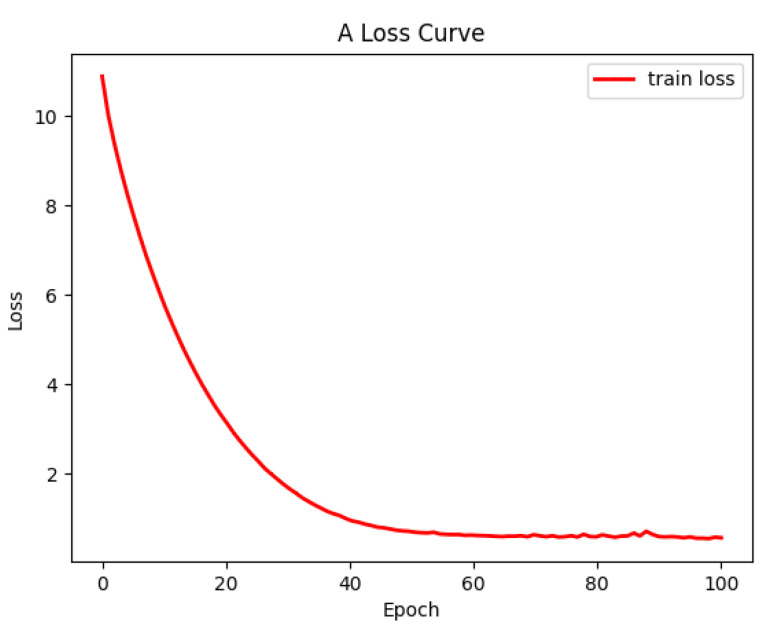
Loss value curve.

**Figure 6 sensors-22-05920-f006:**
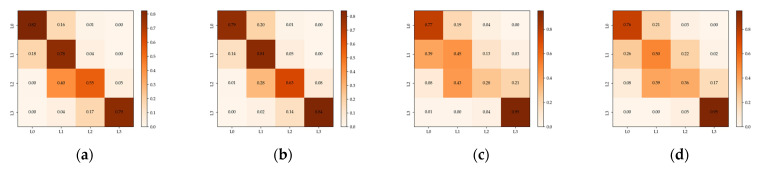
Confusion matrix in Group 3: (**a**) baseline in the Ludian dataset; (**b**) EBDC-Net in the Ludian dataset; (**c**) baseline in the Yushu dataset; (**d**) EBDC-Net in the Yushu dataset.

**Figure 7 sensors-22-05920-f007:**
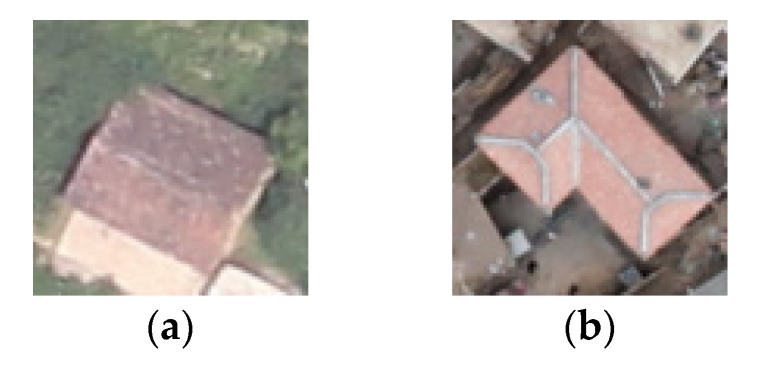
Examples of buildings in different areas: (**a**) Ludian dataset; (**b**) Yushu dataset.

**Figure 8 sensors-22-05920-f008:**
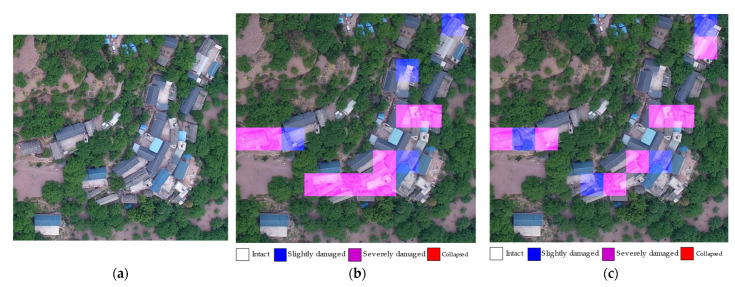
Example of assessment results of building damage in the Yangbi dataset: (**a**) original image; (**b**) EBDC-Net assessment results; (**c**) visual interpretation results.

**Table 1 sensors-22-05920-t001:** Building examples and image information of the four damage levels in the datasets.

Dataset	Intact	Slightly Damaged	Severely Damaged	Collapsed	Image Size	Resolution
Yushu Dataset [16]	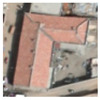	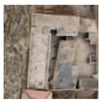	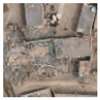	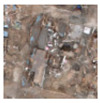	88 × 88 Pixel	0.1 m
Ludian Dataset [16]	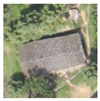	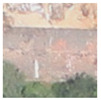	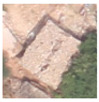	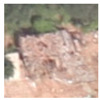	88 × 88 Pixel	0.2 m
Yangbi Dataset	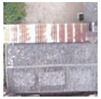	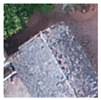	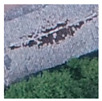	-	88 × 88 Pixel	0.03–0.2 m

**Table 2 sensors-22-05920-t002:** Statistics on the number of buildings sampled in the datasets with the four levels of damage.

Damage Level	Description	Ludian Dataset	Yushu Dataset	Yangbi Dataset
L0	Intact	1630	778	928
L1	Slightly damaged	3074	918	202
L2	Severely damaged	1685	665	111
L3	Collapsed	1984	1140	-
Total		8337	3510	1241

**Table 3 sensors-22-05920-t003:** Distribution of building damage levels for the three classification criteria.

Group	Description	Damage Level
Group 1	Non-collapsed	L0, L1, L2
	Collapsed	L3
Group 2	Intact	L0, L1
	Severely damaged	L2
	Collapse	L3
Group 3	Intact	L0
	Slightly damaged	L1
	Severely damaged	L2
	Collapse	L3

**Table 4 sensors-22-05920-t004:** The effects of different batch sizes on model classification accuracy.

Test	OA (%)	Kappa	MSE
LR-0001-BS-8	68.39	0.56	0.44
LR-0001-BS-16	75.00	0.65	0.28
LR-0001-BS-32	77.49	0.69	0.26
LR-0001-BS-64	75.96	0.67	0.28

**Table 5 sensors-22-05920-t005:** The effects of different learning rates on model classification accuracy.

Test	OA (%)	Kappa	MSE
LR-001-BS-32	29.02	0.06	0.93
LR-0001-BS-32	77.49	0.69	0.26
LR-00001-BS-32	68.58	0.57	0.40

**Table 6 sensors-22-05920-t006:** The quantitative comparison of different baseline models for classification of building damage in the Ludian dataset.

Model Name	Group 1			Group 2			Group 3		
	OA (%)	Kappa	MSE	OA (%)	Kappa	MSE	OA (%)	Kappa	MSE
DenseNet	91.09	0.76	0.09	79.21	0.65	0.27	69.63	0.58	0.43
ResNet50	92.52	0.79	0.07	81.32	0.68	0.20	71.26	0.61	0.37
InceptionV3	92.62	0.79	0.07	81.41	0.68	0.23	72.99	0.63	0.35
Xception	92.43	0.79	0.08	79.12	0.64	0.26	65.80	0.53	0.43
MobileNet	92.81	0.80	0.07	80.46	0.66	0.24	72.22	0.62	0.33
VGG16	93.29	0.81	0.06	81.61	0.68	0.22	73.37	0.63	0.30
Baseline	**93.39**	**0.82**	**0.06**	**83.52**	**0.72**	**0.18**	**74.23**	**0.64**	**0.30**

The bold font indicates the best accuracy of each indicator.

**Table 7 sensors-22-05920-t007:** The quantitative comparison of different baseline models for classification of building damage in the Yushu dataset.

Model Name	Group 1			Group 2			Group 3		
	OA (%)	Kappa	MSE	OA (%)	Kappa	MSE	OA (%)	Kappa	MSE
DenseNet	92.44	0.82	0.07	76.32	0.60	0.38	63.34	0.50	0.66
ResNet50	93.58	0.85	0.06	76.03	0.61	0.32	63.20	0.50	0.60
InceptionV3	92.58	0.83	0.07	75.46	0.58	0.35	63.62	0.51	0.58
Xception	92.86	0.84	0.07	76.46	0.61	0.29	61.91	0.48	0.59
MobileNet	92.87	0.84	0.07	75.19	0.58	0.39	64.05	0.51	0.59
VGG16	**93.72**	**0.86**	**0.06**	76.18	0.63	0.27	63.91	0.51	0.52
Baseline	93.30	0.85	0.07	**77.03**	**0.64**	**0.26**	**65.19**	**0.53**	**0.47**

The bold font indicates the best accuracy of each indicator.

**Table 8 sensors-22-05920-t008:** The effects of different modules in EBDC-Net on the accuracy of building damage classification in the Ludian dataset.

Model Name	Group 1			Group 2			Group 3		
	OA (%)	Kappa	MSE	OA (%)	Kappa	MSE	OA (%)	Kappa	MSE
Baseline	93.39	0.82	0.07	83.52	0.72	0.18	74.23	0.64	0.30
Baseline +R	93.58	0.82	0.06	83.81	0.72	0.18	75.19	0.66	0.27
Baseline +R+P	93.67	0.82	0.06	84.10	0.72	0.17	76.14	0.67	0.27
Baseline +R+P+C	**94.44**	**0.83**	**0.06**	**85.53**	**0.75**	**0.17**	**77.49**	**0.69**	**0.26**

The bold font indicates the best accuracy of each indicator.

**Table 9 sensors-22-05920-t009:** The effects of different modules in EBDC-Net on the accuracy of building damage classification in the Yushu dataset.

Model Name	Group 1			Group 2			Group 3		
	OA (%)	Kappa	MSE	OA (%)	Kappa	MSE	OA (%)	Kappa	MSE
Baseline	93.30	0.85	0.07	77.03	0.64	0.26	65.19	0.53	0.47
Baseline +R	93.58	0.86	0.06	78.60	0.64	0.27	65.48	0.53	0.47
Baseline +R+P	93.86	0.86	0.06	78.74	0.65	0.24	66.48	0.66	0.45
Baseline +R+P+C	**94.72**	**0.88**	**0.05**	**79.02**	**0.65**	**0.26**	**67.62**	**0.56**	**0.42**

The bold font indicates the best accuracy of each indicator.

**Table 10 sensors-22-05920-t010:** Comparison of different methods for the task of classifying building damage in the Ludian dataset.

Model Name	Group 1			Group 2			Group 3		
	OA (%)	Kappa	MSE	OA (%)	Kappa	MSE	OA (%)	Kappa	MSE
Res-CNN [11]	89.27	0.68	0.10	75.86	0.54	0.37	63.99	0.50	0.49
Dense-CNN [13]	89.08	0.71	0.11	77.68	0.61	0.31	69.15	0.57	0.38
VGG-GAP [14]	93.30	0.81	0.06	83.14	0.71	0.19	73.27	0.64	0.32
VGG-OR [16]	93.29	0.81	0.06	84.58	0.74	0.17	75.57	0.66	0.26
EBDC-Net	**94.44**	**0.83**	**0.06**	**85.53**	**0.75**	**0.17**	**77.49**	**0.69**	**0.26**

The bold font indicates the best accuracy of each indicator.

**Table 11 sensors-22-05920-t011:** Comparison of different methods for the task of classifying building damage in the Yushu dataset.

Model Name	Group 1			Group 2			Group 3		
	OA (%)	Kappa	MSE	OA (%)	Kappa	MSE	OA (%)	Kappa	MSE
Res-CNN [11]	91.44	0.91	0.09	75.32	0.58	0.37	58.20	0.43	0.64
Dense-CNN [13]	92.15	0.83	0.08	76.03	0.60	0.32	59.49	0.44	0.74
VGG-GAP [14]	94.00	0.87	0.06	78.89	0.67	0.23	63.77	0.51	0.56
VGG-OR [16]	93.30	0.85	0.07	77.75	0.65	0.26	65.05	0.53	0.43
EBDC-Net	**94.72**	**0.88**	**0.05**	**79.02**	**0.65**	**0.26**	**67.62**	**0.56**	**0.42**

The bold font indicates the best accuracy of each indicator.

**Table 12 sensors-22-05920-t012:** Comparison of different methods in the refined assessment of building damage (L0: intact; L1: slightly damaged; L2: severely damaged; L3: collapsed).

Classification Results	Images
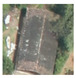	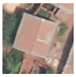	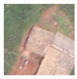	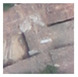	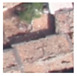	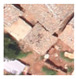	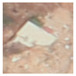	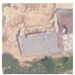
Ground Truth	L0	L0	L1	L1	L2	L2	L3	L3
Dense-CNN	L1	L1	L2	L2	L2	L3	L3	L3
VGG-GAP	L0	L2	L2	L2	L3	L2	L3	L3
VGG-OR	L0	L1	L2	L2	L2	L2	L2	L2
EBDC-Net	L0	L0	L1	L1	L2	L2	L3	L3

**Table 13 sensors-22-05920-t013:** Impact of historical data on the refined assessment of building damage.

Test Name	Group 1			Group 2			Group 3		
	OA (%)	Kappa	MSE	OA (%)	Kappa	MSE	OA (%)	Kappa	MSE
Test1	88.30	0.75	0.12	69.19	0.52	0.39	56.63	0.41	0.76
Test 2	94.72	0.88	0.05	79.02	0.65	0.26	67.62	0.56	0.42
Test 3	**95.86**	**0.91**	**0.04**	**80.02**	**0.68**	**0.22**	**68.33**	**0.57**	**0.43**

The bold font indicates the best accuracy of each indicator.

## Data Availability

The Ludian and Yushu datasets are freely available online, and can be found at https://github.com/city292/build_assessment (accessed on 1 April 2022).

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
