# Peer review of "Classification of Building Damage Using a Novel Convolutional Neural Network Based on Post-Disaster Aerial Images"

_sensors, 2022, doi:10.3390/s22155920_

Round 1
Reviewer 1 Report
This study proposes a CNN-based network for the finer classification of building damage. It is essential for the rapid identification of disaster building damage. The authors used three datasets to conduct experiments to verify the effectiveness of the method. The experiments are relatively adequate, but some information is missing or the results are not well presented.
Comments:
1. How was the data in Table 1 obtained and produced, and can it be made public?
2. Table 3: Why are they divided into three groups? (The combination of Intact, Slightly damaged, Collapse is missing)
3. Implementation details of the experiment: How are the parameters of the comparison method set? How to ensure that the parameters of other methods are also optimal?
4. Tables 4, 5 and 6: There are three datasets, but only the Ludian dataset is shown here. Results of other areas also need to be listed. In addition, confusion matrix analysis is also important.
5. Figures 6 and 7: The corresponding quantitative evaluation results are missing. With this sampling strategy, if a house is cropped into two or more small patches, can it be correctly identified? How to solve this problem? In my opinion, this issue should be discussed carefully, as it determines whether the proposed algorithm is suitable for practical applications.
6. The conclusion section repeats information from other parts of the manuscript and should be reorganized. It should contain conclusive content.
Author Response
July 23, 2022
Dear Reviewer,
Thank you for your time and efforts on reviewing the paper. According to these valuable comments, we have carefully revised our manuscript “Classification of Building Damage Using a Novel Convolutional Neural Network Based on Post-Disaster Aerial Images”, and a point-by-point response to the comments is attached to this letter. The changes have been highlighted by colored text in the revised paper. For detailed changes, please refer to the response letter.
Many thanks for your considerations.
Best Regards,
Dr. Zhonghua Hong
Shanghai Ocean University

Reviewer 2 Report
[Comment 1] Novelty
[Subcomment 1a] The authors should compare the performance of their proposed method when using only two categories as well, by comparing it with the state-of-the-art methods (the ones listed in the literature review section).
[Subcomment 1b] In lines 45-46, the authors stated that they are not using pre-disaster data. But, using historical data term looks similar with the pre-disaster data. What is meant by the historical data?
[Comment 2] Methodology
[Subcomment 2a] Do the authors obtain the result of the "feature extraction encoder network" before proceeding to the "building damage classification network"? If yes, I suggest the authors provide examples of the data to provide more clarity.
[Subcomment 2b] (lines 237-238) Please justify why the authors use such parameter values by providing a reference or explaining about the parameter tuning/testing procedure.
[Comment 3] Reference
[Subcomment 3a] (lines 102-103) Please share the reference for this data as well.
[Subcomment 3b] I suggest the authors upload the data (that are classified by the authors into 2, 3, then 4 classes) onto an online repository and share the link in the manuscript to allow other researchers conduct next studies as well.
[Comment 4] Writing quality and clarity
(Figure 6) The color cannot be seen well because of the background color. I suggest the authors place a clean figure at the left side, then a marked image at the right side with better color contrast or another marking method.

Author Response

(The authors gave the same response as above.)

Round 2
Reviewer 1 Report
Thank you very much for the replies to my concerns. The authors responded to comments 3 and 5, but I suggest that authors add relevant responses to the main text, as this information is also of interest to readers. Also, the English language needs to be polished and there are errors in the text.
1. Lines 277: “Table 6 and 7 shows” should be “show”.
2. The legend of Figure 7 needs to be enlarged.
Author Response
Many thanks for your careful reviewing on our paper. The paper has been carefully revised according to your comments and suggestions, which are very valuable to the improvement of this paper. The following is a summary on how the paper is revised according to your comments.

Reviewer 2 Report
The authors must add Tables 1 and 2 and Figure 1 provided by the authors (related to the parameter tuning) in the response to authors file into the manuscript.
Author Response
Many thanks for your careful reviewing on our paper. The paper has been carefully revised according to your comments and suggestions, which are very valuable to the improvement of this paper. Attached file is a summary on how the paper is revised according to your comments.
